# Enablers and barriers to treatment adherence in heterozygous familial hypercholesterolaemia: a qualitative evidence synthesis

Fiona J Kinnear,[1] Elaine Wainwright,[2,3] Rachel Perry,[1] Fiona E Lithander,[1] Graham Bayly,[4] Alyson Huntley,[5] Jennifer Cox,[1] Julian PH Shield,[1] Aidan Searle[1]

For numbered affiliations see end of article.

**Correspondence to**
Fiona J Kinnear;
fiona.kinnear@bristol.ac.uk

## ABSTRACT

**Objectives** Individuals with heterozygous familial hypercholesterolaemia (FH) are at high risk of developing cardiovascular disease (CVD). This risk can be substantially reduced with lifelong pharmacological and lifestyle treatment; however, research suggests adherence is poor. We synthesised the qualitative research to identify enablers and barriers to treatment adherence.

**Design** This study conducted a thematic synthesis of qualitative studies.

**Data sources** MEDLINE, Embase, PsycINFO via OVID, Cochrane library and CINAHL databases and grey literature sources were searched through September 2018.

**Eligibility criteria** We included studies conducted in individuals with FH, and their family members, which reported primary qualitative data regarding their experiences of and beliefs about their condition and its treatment.

**Data extraction and synthesis** Quality assessment was undertaken using the Critical Appraisal Skills Programme for qualitative studies. A thematic synthesis was conducted to uncover descriptive and generate analytical themes. These findings were then used to identify enablers and barriers to treatment adherence for application in clinical practice.

**Results** 24 papers reporting the findings of 15 population samples (264 individuals with FH and 13 of their family members) across 8 countries were included. Data captured within 20 descriptive themes were considered in relation to treatment adherence and 6 analytical themes were generated: risk assessment; perceived personal control of health; disease identity; family influence; informed decision-making; and incorporating treatment into daily life. These findings were used to identify seven enablers (eg, 'commencement of treatment from a young age') and six barriers (eg, 'incorrect and/or inadequate knowledge of treatment advice') to treatment adherence. There were insufficient data to explore if the findings differed between adults and children.

**Conclusions** The findings reveal several enablers and barriers to treatment adherence in individuals with FH. These could be used in clinical practice to facilitate optimal adherence to lifelong treatment thereby minimising the risk of CVD in this vulnerable population.

**PROSPERO registration number** CRD42018085946.

## Strengths and limitations of this study

► This is the first thematic synthesis of the qualitative literature exploring the beliefs and experiences of individuals with familial hypercholesterolaemia to identify enablers and barriers to treatment adherence that can be targeted in clinical practice.

► Robust procedures for conducting a thematic synthesis were adopted, informed by the Cochrane Qualitative Research Methods Group guidelines and they were reported in line with the Enhancing Transparency in Reporting the Synthesis of Qualitative Research statement.

► The barriers and enablers were identified from themes which were representative of all the included studies, increasing their validity.

► While included studies were conducted across eight countries, all were within the developed world which could limit the generalisability of the findings.

## INTRODUCTION

Heterozygous familial hypercholesterolaemia (FH) is one of the most common inherited genetic disorders, estimated to affect as many as 1 in 250 individuals worldwide.[1 2] Left untreated, the exposure to chronically elevated levels of low density lipoprotein cholesterol (LDL-C) from birth confers an increased risk of cardiovascular disease (CVD),[2 3] with approximately 50% and 85% of affected women and men, respectively, experiencing a coronary event before the age of 65.[4] While this risk can be significantly reduced with early detection and treatment, many affected individuals remain at higher risk of premature CVD morbidity and mortality.[5–9] The most beneficial effects of treatment are evident in primary prevention before the onset of CVD.[5 10] With diagnostic rates as low as 1% in some countries,[11] current efforts are focused on identifying individuals with FH via screening and genetic testing programmes.[12 13] Treated as

outpatients and asked to follow lifelong treatment, it is critical to ensure that this increasing patient group are able to self-manage their disease. With many patients not reaching treatment targets,[14–16] it is an area that warrants further investigation.

To improve adherence to treatment recommendations, an understanding of the factors affecting adherence is required. The American Heart Association has recognised the need to gain a deeper understanding of the experiences of individuals with FH before addressing the further identified research gaps.[17] Preliminary research has found the beliefs and attitudes of patients with FH towards the recommended treatment exert a significant effect on their intention to engage in these behaviours.[18 19] Qualitative research can provide further insight to how these beliefs and attitudes are developed and the nature by which they may influence subsequent behaviours.[20] Its exploratory nature also allows for the identification of other factors influencing an individual's ability and motivation to comply with treatment.[21 22]

Qualitative research conducted in patients with FH has found illness knowledge,[23] risk perception,[24] a lack of symptoms[25] and family history of disease[26] to influence treatment adherence. However, the transferability of these findings beyond the sample they are conducted in is limited.[27] Qualitative syntheses, which bring together the findings from individual qualitative studies, can be used to gain a more in-depth understanding of the issue and identify common themes which are applicable to a wider range of contexts.[28 29] It is recognised as an important source of evidence to inform healthcare interventions and policy development[30–32] including those targeting treatment adherence[33–35] and is advocated by the World Health Organisation (WHO) and the Cochrane Collaboration Group.[28 36] Given the limited literature concerning treatment adherence in FH, the results of this synthesis will also be compared with the results of research investigating treatment adherence in similar medical conditions.

### Objectives

1. Identify how the experiences and beliefs of individuals with FH influence their adherence to pharmacological and lifestyle treatment recommendations.
2. Explore if these findings differ between children and adults.
3. Use the findings to generate new understandings of the enablers and barriers to treatment adherence to inform clinical practice.

### MATERIALS AND METHODS

The methods used for this qualitative synthesis are briefly described below with full details available in the published protocol[37] and on the PROSPERO database (registration number CRD42018085946). Minor deviations to the protocol were made, outlined in online supplementary file 1. The Enhancing Transparency of Reporting the synthesis of Qualitative research (ENTREQ) statement[38] has been followed and a checklist is available in online supplementary file 2.

### Search strategy

A comprehensive, systematic and preplanned search was conducted to find all available qualitative evidence—full details are available in online supplementary file 3.

### Selection criteria

*Participants:* Individuals with a clinical or genetic diagnosis of heterozygous FH. No restrictions were placed on age or history or CVD. Individuals with homozygous FH were not included.

*Phenomena of interest:* The experiences and beliefs of individuals with FH, and their family members, regarding their condition, its long-term health consequences and recommended pharmacological and lifestyle change treatment.

*Types of studies:* Only papers reporting primary qualitative data were included. Questionnaire studies were not included. Papers reporting both quantitative and qualitative data were included if the qualitative data could be independently extracted. Multiple papers reporting findings from the same sample of participants were included if they reported unique data.

*Intervention/exposure:* Treatment was defined as any behavioural action undertaken by an individual in an effort to manage his/her FH diagnosis.

*Setting:* No restrictions were placed on the country in which study was conduction, nor the location at which data were collected from individuals.

### Quality appraisal

The methodological quality of the studies was assessed using the Critical Appraisal Skills Programme (CASP) tool for reviewing qualitative research.[39] As the purpose of the quality appraisal was to determine the methodological strengths and limitations of studies included in the synthesis, the lead authors of each paper were contacted to obtain further information in an attempt to overcome the recognised issued of poor reporting in qualitative research. Full details of how this tool was used are available in online supplementary file 4.

### Data extraction

Methodological and contextual information from each paper were extracted into a table designed for this review by two reviewers independently (FK, JC) after piloting in five papers. Two reviewers (FK, AS) independently reviewed all text under the results, conclusions and discussion headings of all papers, as well as any supplementary files. Any data identified to be relevant to the research questions were extracted electronically using a tool designed for this review. In instances in which multiple papers reported the findings from a single study, data from the primary paper PhD theses were extracted first, before supplementary publications were reviewed for any additional, unique data. Results were compared and discussed until agreement was reached.

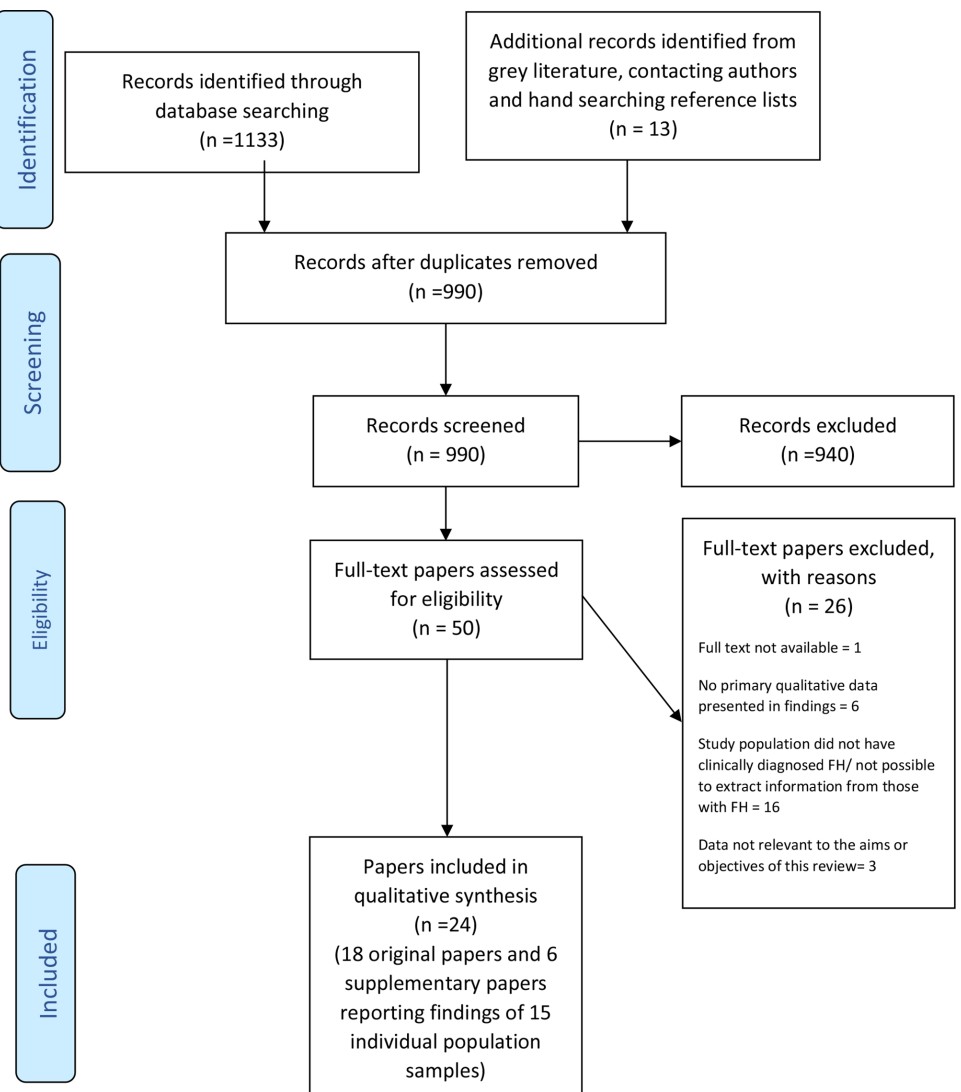

**Figure 1** Preferred Reporting Items for Systematic Reviews and Meta-Analyses flow diagram. FH, familial hypercholesterolaemia.

## Data analysis

Thematic synthesis,[40] a widely accepted and commonly used approach in qualitative syntheses, was used.[41 42] It involved three stages: line by line coding of the extracted data, generation of descriptive themes and development of analytical themes. Using NVivo software, two reviewers (FK, AS) carried out the coding independently. The subsequent stages were carried out collaboratively between three reviewers (FK, AS, EW). To enhance transparency, full details are available in online supplementary file 5. The findings were discussed with three clinicians (JPHS, GB, PD) currently providing care to individuals with FH to help develop feasible and relevant recommendations for clinical practice.

## Sensitivity analysis

To ensure the quality appraisal results were used in a meaningful way,[41 43] post-hoc sensitivity analysis was carried out by three reviewers (FK, AS, EW) to examine the extent to which the synthesis results were affected by

exclusion of poor quality papers, described in full elsewhere.[44] It involved examining if any themes were lost when each paper was removed from synthesis and evaluate if there was a significant impact on the 'thickness' of findings reported within each theme. 'Thickness' refers to the depth, scope and context of findings which could influence the transferability and credibility of the results to the wider FH patient population.[45] This was carried out through discussion between three reviewers (FK, AS, EW).

## Patient and public involvement

Patients or members of the public were not involved in this study.

## RESULTS

The titles and abstracts of 990 unique citations identified by the searches were screened, with 50 progressing to screening at the full-text level. Twenty-six papers

were excluded at this stage due to the full text not being available (n=1), no primary qualitative data being presented in the findings (n=6), the study population not having a clinical diagnosis of FH or inability to selectively extract data from those with a diagnosis in a mixed population (n=16) and data not being relevant to the aims of this review (n=3). Multiple papers reporting findings from the same sample of individuals and three PhD papers,[46–48] two of which had supplementary papers published in addition to the originally reported theses, were included. Each paper was considered to be a separate primary paper and referenced separately. In total, 24 papers were included in the synthesis, comprising 18 original[23 25 46–61] and 6 supplementary papers[24 26 62–65] reporting the findings of 15 population samples (figure 1).

## Characteristics of studies and participants

In total, 264 individuals with FH and 13 family members were involved, aged 8–69 years. Seven papers[24 25 46 58 59 62 63] reported findings from three samples which included individuals under 18 years. Four papers reported parental views of having children with FH.[25 56 58 59] Full characteristics of the included papers and samples are presented in table 1.

## Quality appraisal and sensitivity analysis

Appraisal scores of papers ranged from 11 to 20 out of 20, with 11 rated high, 7 medium and 6 low (table 1). The most common methodological limitations uncovered were relating to ethical issues, researcher reflexivity and rigour of data analysis. Consideration of a researcher's potential influence and bias on data collection and analysis was critically examined fully in 7 papers,[24 25 46 47 58 62 63] partially in 10[23 26 48 50 51 55 57 59 64 65] and not addressed in 7.[49 52–54 56 60 61] Ethical approval was obtained, or reasons given for exemption, in all but two papers[60 61]; however, participants were not provided adequate information about withdrawal and anonymisation of data processes in a further four papers.[25 49 50 58] The data analysis was carried out by one researcher only in seven papers[23 26 47 48 56 64 65] and it was unclear if more than one person was involved at each stage of analysis in four papers.[51 52 60 61]

Eight lead authors responded to our request for further information, providing information for 16 of the 24 papers. Five of the six papers rated as low quality were papers for which the author did not respond. This reflects our belief that low ratings may be reflective of poor reporting rather than poor methodology, supporting our decision not to exclude papers. The sensitivity analysis carried out found that the removal of the six poor quality papers had no significant effect on the synthesis findings—in both the descriptive and analytical themes uncovered and the depth of the findings. More detailed information of methodological and transferability issues is available in online supplementary file 4.

## Data analysis

Six analytical themes were derived from the findings captured by 20 identified descriptive themes, as displayed in table 2 alongside illustrative quotes. Table 3 shows the occurrence of the descriptive themes within the extracted data from the 24 papers. While each analytical theme has a direct influence on treatment adherence, they are not exclusive in nature and inter-theme relationships are evident as displayed in the thematic schema in figure 2. Additionally, some themes by their integrative nature had a greater influence on treatment adherence as indicated by the shaded boxes. There were insufficient data regarding children and young people to explore whether the findings differed from adults.

Seven enablers and six barriers to treatment adherence (table 4) were uncovered during the analysis of these themes and are described alongside the analytical themes below. In this section 'treatment' refers to both lifestyle and medication behaviours, unless otherwise specified.

## Analytical themes

### Risk assessment

Individuals lived experience of their disease, coupled with their beliefs concerning its known risks, increased or decreased their sense of vulnerability to its long-term health consequences. Knowledge of how FH had affected family members was the most prevalent factor considered by individuals when assessing their risk. Individuals with lived experience of a family member being ill or dying prematurely due to FH had a heightened sense of risk.[46 48 49 52 55 56 58–62] Individuals unaware of FH in their families or with family members living a life unaffected by its consequences perceived themselves at lower risk[46 52 56 58 61 62]: 'My dad's now in his 70s…it's not something I feel particularly threatened about having.'[56]

As FH does not 'make you feel ill',[52] individuals found having FH 'easy to forget, and easy not to take seriously.'[47] This was salient among younger individuals without existing CVD symptoms[23 25 47 48 58 59 65] for whom '…cholesterol always comes last. It will never be a focus until something happens to me.'[47] Older individuals who had lived through, or were currently experiencing CVD, perceived themselves at higher risk.[23 56 61 62] Others framed their perception of risk in the context of the risk they believed other diseases presented, concluding that FH health consequences were not as serious[23 47 48 51 53 54 61]: 'I didn't think it was life threatening, like being told you've got cancer.'[23]

For the majority of individuals, their risk assessment led to a perception that FH did not present a great risk to their current or long-term health.[23 47–49 51 56 59–61] This mismatch between the perceived and actual risk has been identified as a barrier to treatment adherence.

### Perceived personal control of health

Individuals acknowledged the threat that FH posed to their health, but there was a widely held belief that they had the ability to modify their own personal risk.[24 47 49 51 53–62]

**Table 1** Characteristics of included studies

| Sample, number | Author and date of paper | CASP quality rating*† | Research aim | Country | Recruitment setting | Sample size‡ | Sample characteristics | Data collection methods |
|---|---|---|---|---|---|---|---|---|
| 1 | Agard et al, 2005[49] | Low | To explore the extent to which FH influences the life of the patients affected | Sweden | Outpatients treated at lipid clinic | 23 | 10 M and 13 F; Mean age 48 years (range 31–67 years); 4 with or had Hx of CVD | Face to face SSI |
| 2 | DeAngelis et al, 2017[50] | Low | To determine individual and group patient ideas and priorities regarding ways to enhance their own health | USA | Patients and family from patient centred outcomes research institute and outpatient clinic | 7 | 6 patients with FH, 1 family member | 15 group meetings |
| 3 | Frich, 2007[46] | High* | To explore how individuals with FH perceive and manage their condition | Norway | Specialist clinic for metabolic lipid disorders | 40 | 20 M and 20 F; Mean age 31 years (range 14–57 years); 7 had CVD symptoms; 19 had children | Face to face SSI |
| | Frich et al, 2006[62] | High* | To explore how patients with diagnosis of FH understand and perceive their vulnerability to CHD | | | | | |
| | Frich et al, 2007[63] | High* | To explore how patients at risk of CHD portray candidates for CHD | | | | | |
| | Frich et al, 2007[24] | High* | To explore patients' experiences of guilt and shame with regard to how they manage FH | | | | | |
| 4 | Hallowell et al, 2017[51] | High* | To investigate index patients' experiences of undergoing DNA testing as part of screening programme | Scotland | Two lipid clinics | 38 | 17 M and 21 F; Mean age 52.6 years (range 18–67 years); 31 had children; 16 educated to university level | Face to face in-depth interviews, (one online) |
| | Jenkins et al, 2013[53] | Medium* | To explore patients' interpretations of their DNA results for FH | | | | | |
| | Jenkins et al, 2013[52] | Low* | To explore the concept of inter-embodiment and its potential for advancing sociological research into illness biography and genetic identity | | | | | |

Continued

**Table 1** Continued

| Sample, number | Author and date of paper | CASP quality rating*† | Research aim | Country | Recruitment setting | Sample size‡ | Sample characteristics | Data collection methods |
|---|---|---|---|---|---|---|---|---|
| 5 | Hardcastle et al, 2015[23] | High* | To investigate the perceptions and experiences of patients with a genetic diagnosis of FH involved in a cascade screening programme. To explore how these patients conceptualise FH and how such beliefs affect treatment compliance and lifestyle changes | Australia | Lipid disorders clinic | 18 | 10 M and 8 F; Mean age 50.2 years (range 25–74 years); 2 had CVD symptoms | Face to face SSI |
| 6 | Hollands et al, 2012[54] | Low | Examine the impact of disease risk assessments based on both genetic and non-genetic information, or solely non-genetic information | UK | Lipid clinics at 11 hospitals | 20 | 12 M and 8 F; Mean age 30.9 years for DNA diagnosed and 40.7 years for non-DNA; 17 white, 1 white Asian, 2 black Caribbean | Three telephone interviews |
| 7 | Hollman et al, 2004[55] | High* | To describe the QOL and to understand the underlying meaning of the concept of QOL in patients with FH | Sweden | Outpatient clinic | 12 | 6 M and 6 F; 20–69 years; 7 had children; 3 university level education; no Hx of CHV | Face to face SSI |
| 8 | Keenan et al, 2018[56] | Medium* | To explore parent's views and experiences of genetic testing and early treatment of children with FH in Scotland, experiences of their children's care pathway and to identify any barriers or facilitators in testing and treatment uptake | Scotland | Clinical genetic services and lipid clinics from 3 sites | 17 | 6 M and 11 F; 20–69 years; all white; 12 had post-secondary qualifications; 3 symptoms or Hx of CVD | SSI (15 face to face, 2 over phone) |
| 9 | Kirkegaard et al, 2014[57] | Medium* | Explore how cholesterol reducing medication and risk of CVD are interpreted by asymptomatic patients with high cholesterol | Denmark | 5 GP centres | 3 | 1 M and 2 F; 24–62 years; no CVD symptoms | Face to face SSI |

Continued

**Table 1** Continued

| Sample, number | Author and date of paper | CASP quality rating*† | Research aim | Country | Recruitment setting | Sample size‡ | Sample characteristics | Data collection methods |
|---|---|---|---|---|---|---|---|---|
| 10 | Mackie et al, 2015[58] | High | Explore how family medical history, family narratives of medical experiences and AYA medical experiences together function as 'experiential evidence' and influence screening and treatment decisions | USA | Paediatric preventative cardiology practice | 24 | 12 AYAs with FH and 12 parents of AYAs with FH (four dyads) AYAs:6 M and 6 F; Mean age 18.4 years; 9 white, 1 black and 1 Asian Parents: 2 M and 10 F; Mean age 49.3 years; 1 Asian, 9 white | Face to face SSI with AYA and parent separately |
| | Sliwinski et al, 2017[25] | High | To examine challenges transitioning to adult care for young adults with FH, and their parents, in the context of 2 developmental tasks: transitioning from childhood to early adulthood and summing responsibility for self-management of a chronic disease | | | | | |
| 11 | Meulenkamp et al, 2008[59] | High* | To study the experiences of children identified by family screening who were found to be a mutation carrier for a genetic CVD | Netherlands | Paediatric lipid clinic | 16 children from 10 families | 5 M and 11 F; 8–17 years Number and age of parents not given | Face to face SSI (children and parents separately) |
| 12 | Mortensen et al, 2008[60] | Low | Comparative study to examine the QOL impact of FH in patients who had and had not reached the target of treatment | Denmark | Centre of inherited CVD | 10 | 6 M and 4 F; 20–72 years; no CVD Hx | Focus groups |
| 13 | Urke, 2016[47] | High | Explore how young adults, who stopped attending lipid clinic for medical and nutritional consultations, managed challenges related to living with FH and to the lifelong treatment | Norway | Outpatient clinic | 11 | 6 M and 5 F; Median age 29 years (range 26–35 years); 8 educated to university levels | SSI (9 face to face 2 over phone) |

Continued

**Table 1** Continued

| Sample, number | Author and date of paper | CASP quality rating*† | Research aim | Country | Recruitment setting | Sample size‡ | Sample characteristics | Data collection methods |
|---|---|---|---|---|---|---|---|---|
| 14 | Weiner, 2006[48] | High * | How much and in which way patients with FH and professionals involved with the condition construct FH and CHD as genetic conditions | England | Lipid clinic | 31 | 17 M and 14 F; Mean age 52 years (range 24–69 years); 31 white; 15 with current CVD | Face to face SSI |
| | Weiner and Durrington, 2008[26] | Medium* | To explore patients' understanding and experiences of FH and the significance of the hereditary aspect of the condition | | | | | |
| | Weiner, 2009[64] | Medium* | Consider how people with FH construct FH, high cholesterol and CHD | | | | | |
| | Weiner, 2011[65] | Medium* | Explore the notion of genetic responsibility, focussing particularly on responsibilities to family and kin | | | | | |
| 15 | Senior et al, 2002[61] | Low | Investigate perceptions of having an inherited predisposition to heart disease in people diagnosed with, and receiving treatment for FH | England | 2 lipid clinics | 7 | 5 M and 2 F; 39–58 years | Face to face SSI |

*CASP score: high=18–20; medium=14–17; low quality=<14.
†Papers for which lead author provided requested further information are marked with *.
‡The sample size and characteristics describe only those in sample with clinically diagnosed heterozygous FH and their family members.
AYA, adolescent and young adult; CHD, coronary heart disease; CVD, cardiovascular disease; F, female; Hx, history; M, male; QOL, quality of life; SSI, semistructured interview.

**Table 2** Analytical themes and their composite descriptive themes with illustrative quotes

| Analytical theme | Descriptive themes | Illustrative quotes from participants (first order) | Illustrative interpretations from authors (second order) |
|---|---|---|---|
| Risk assessment | FH is a silent disease | 'Not a condition that has any symptoms, that makes you feel ill or anything.'[52] | 'The majority of interviewees did not look on the condition as a disease…If they were not affected by a cardiac disease… they regarded themselves as healthy.'[49] |
| | Family history modifies perception of FH-related threat to health | 'I'm not going to get past sixty. Dad never got past sixty.'[53] | 'To them, reaching the age of death of a parent with FH was anticipated with fear of having a heart attack themselves.'[60] |
| | FH is not as threatening to health as other conditions | 'Its not that bad….Its not like having something like Huntington's or something like that.'[51] | They mentioned conditions with more drastic consequences such as allergies, epilepsy or diabetes.'[47] |
| Perceived personal control of health | FH is a manageable condition | 'Well it's treatable isn't it by diet and drugs. It's not something that's incurable.'[48] | 'FH carrier children demonstrated high feelings of control over their condition.'[59] |
| | Individuals feel personally responsible for managing their FH | 'It means you could be in danger of like what could possibly happen like in the future if you don't change anything.'[58] | 'FH patients have a strong desire to empower themselves in order to improve their own health.'[50] |
| | FH medication is effective | 'I believe that as I am taking the pills that my risk of heart attack is no greater than anyone else of my age or weight.'[61] | 'Preventative medical treatment built confidence in the potential for living a long life.'[55] |
| | FH lifestyle treatment viewed as less important than medication | 'I could never get that down no matter how much dieting or exercise I do…so it can only be reduced through medication.'[48] | 'Many tended to devalue the importance of lifestyle changes in controlling FH and place their hope in medication.'[23] |
| Disease identity | Importance of establishing that high cholesterol levels are not self-inflicted | 'It enables me to emphasise that it is not my fault, that it's something inherited.'[62] | 'They always described FH as a hereditary condition to underline that their cholesterol issues were not due to unhealthy lifestyle.'[60] |
| | Receiving genetic diagnosis provides certainty | 'I guess it is a relief in a funny way because I had an answer to what was quite a surprising medical condition that I had… so at least I know now and can take preventative measures.'[54] | 'It provided an aetiological explanation and diagnostic label, confirmed current risk management practices…'[24] |
| The influence of family | Desire to protect children | 'We want to help him…(so) we have decided to give him statins until he is 16… we've covered him until he's old enough to decide for himself.'[56] | 'In fact, the main concern for the affected parents appeared to be the well-being of their children…'[49] |
| | Parental influence on treatment related behaviours | 'My parents, specifically my mom, were really integral in teaching us types of food to eat.'[25] | 'AYAs expressed how their perceptions of their parents experience have influenced their perceptions of the respective treatment options.'[58] |
| | FH and its treatment become normalised within families | 'Since I grew up with FH and had a relatively good diet and good habits and routines, it makes it easier.'[47] | 'FH carrier children typically reported it had become habit to maintain a healthy, non-fat diet. Commonly the whole family, including the non-carriers, kept to the same diet restrictions.'[59] |
| Informed decision making | HCP interactions | 'My daughter. I don't think she really understood what (high cholesterol) really meant until she came here and talked with doctor.'[58] | 'The doctors presentation of FH, however, influenced all patients perceptions of the risk and severity of the diagnosis.'[60] |
| | Inadequate and/or incorrect knowledge about FH and treatment | 'in the newspapers, the stories that you cut out butter, red meat, etc, and you'll be okay.'[61] | 'Many informants still had unanswered questions or were felt to lack relevant knowledge.'[49] |
| | Concerns about side effects of FH medication | 'Would I be able to have children at all after taking all these medicines for years?'[49] | 'Parents reported having strong concerns about statin treatment in children, not only because of their long-term safety but also potential side effects.'[56] |

Continued

**Table 2** Continued

| Analytical theme | Descriptive themes | Illustrative quotes from participants (first order) | Illustrative interpretations from authors (second order) |
|---|---|---|---|
| Incorporating treatment into daily life | FH and its treatment does not have big impact on life | 'You don't have to plan your life around it. You don't have to wonder, can you have children or not.'[51] | 'FH was not viewed as a significant burden, but more of a lifestyle adjustment, involving a healthy diet, exercise, and statin treatment from an early age.'[56] |
| | Balancing FH treatment with other competing priorities | 'Our two children, who were often ill….My husband…travelled all the time, so I almost had more than I could put up with at that moment.'[62] | 'Young adults also articulated challenges maintaining diet and exercise regimes while adjusting to a new routine and environment at college or in workforce.'[25] |
| | Lifestyle advice treatment is restrictive and difficult to follow | 'I've changed my diet as much as I can… don't want to bother too much and speculate, live an unworthy life and diet at the age of seventy. I'd rather be happy and die when I'm fifty.'[24] | 'Making dietary changes had been the worst aspect of their condition, and this included people who already had CHD.'[62] |
| | Social implications of following FH treatment | 'Some people comment on the things I eat. And then I'm like 'well actually I have to eat this because I've got FH and I have to watch my diet.'[54] | '10 young adults articulated how concern over peers' opinions or overt peer pressure-restricted social activities centred around eating.'[25] |
| | Desire for further support and guidance | I think having the resources (would make it easy to adhere to lifestyle treatment)… like seeing a nutritionist that can give you options….'[25] | '…expressed a desire to be able to access educational resources in one place and for a way to reach out to others who could provide solidarity, comfort and aid with management of FH.'[50] |

AYA, adolescent and young adult; CHD, cardiovascular heart disease; HCP, healthcare professional.

They recognised that this required active engagement with treatment[23–25 47 49–51 53–56 58 61 62] and held themselves accountable for managing their disease[23–25 47–51 53–58 60–62] experiencing a 'bad conscience'[49] and 'guilt'[63] when they did not meet the expectations they had set themselves. Treatment was perceived to be effective[24 47 49 51 53–62] with individuals viewing FH as 'treatable'[48] and 'controllable'.[23] In particular, medication was regarded by individuals to be a mandatory and effective component of treatment.[24 47 49 51 53–62] They believed FH could be 'solved'[59] with medication and lead to achievement of cholesterol levels 'like most people'.[23] While individuals spoke of their efforts to change their lifestyle behaviours,[24 25 47 49 51 53–62] many believed their cholesterol levels would not be 'radically changed'[61] by doing so[47 48 58 60] as 'doesn't matter what I eat or how much exercise I'm still going to have high cholesterol without tablets'.[23]

This confidence in the ability to successfully self-manage their condition was identified as an enabler to treatment adherence. The perceived effectiveness of medication led to a devaluing of the importance of following lifestyle treatment,[23 47 48 57 58 60] and this prioritisation of medication was identified as a barrier to adhering to lifestyle treatment.

### Disease identity

Individuals placed great importance, especially in social situations, to emphasis that they were 'not to blame'[60] for their high cholesterol.[24 26 48 50 51 53 54 57 60 61 63] High cholesterol was associated with unhealthy lifestyles and individuals wished to distance themselves from this negative connotation.[24 48 54 57 60 61 63] A positive genetic test provided 'a definitive',[51] rather than a possible, explanation for their high cholesterol[50 53 54] and positively influenced individuals' perceptions and behaviours.[24 50 51 53 54] If individuals had been following treatment of their own volition before the diagnosis, it helped 'reaffirm their commitment'[53] to treatment.[51 54] If they had been previously unaware of their condition, it prompted them to seek treatment[53 56]: 'I know now and can take preventative measures'.[54] Therefore, receiving a formal diagnosis was identified as an enabler to treatment adherence as being given a medical explanation empowered individuals to take control of their condition through engaging with treatment.

### Family influence

Parents expressed a high level of concern about the well-being of their affected children[25 48 50 51 53 56 58 59] and this parental responsibility to care for children was identified as another enabler of treatment adherence. They assumed responsibility to ensure their children adhered to medical and lifestyle treatment,[25 48 50 51 53 56 58 59] taking action to 'bring them up with healthy eating habits'[51] and 'make sure that they take their medication'.[48] This involvement was reflected in the finding of individuals attributing their current treatment knowledge and behaviours to their parents[47–49]: 'everything I've learnt from home'.[47] Parents also made treatment-related decisions on their behalf[25 48 50 53 58 59] until they were 'old enough to decide.'[56] As such, the early adulthood years presented a challenge for treatment adherence as the young adults transitioned

**Table 3** Occurrence of descriptive themes across the included papers and samples*

| Sample number | Paper | FH is a silent disease | Family history modifies perception of FH related threat to health | FH is not as threatening to health as other conditions | FH is a manageable condition | Individuals feel personally responsible for managing their FH | FH medication is effective | FH lifestyle treatment viewed as less important than medication | Importance of establishing that high cholesterol levels are not self-inflicted | Receiving genetic diagnosis provides certainty | Desire to protect children | Parental influence on treatment related behaviours | FH and its treatment become normalised within families | HCP relationships | Inadequate and/or incorrect knowledge about FH and its treatment | Concerns about side effects of FH medication | FH and its treatment does not have big impact on life | Balancing FH treatment with other competing priorities | Lifestyle advice treatment is restrictive and difficult to follow | Social implications of following FH treatment | Desire for further support and guidance |
|---|---|---|---|---|---|---|---|---|---|---|---|---|---|---|---|---|---|---|---|---|---|
| 1 | Agard et al, 2005[49] | ✓ | ✓ |  | ✓ | ✓ | ✓ |  |  |  | ✓ | ✓ |  | ✓ | ✓ | ✓ | ✓ | ✓ | ✓ | ✓ | ✓ |
| 2 | DeAngelis et al, 2017[50] |  | ✓ |  |  | ✓ |  |  |  | ✓ | ✓ |  |  | ✓ | ✓ |  |  |  |  |  | ✓ |
| 3 | Frich, 2007[46] |  | ✓ |  | ✓ | ✓ | ✓ |  |  |  |  |  |  | ✓ | ✓ |  |  | ✓ |  |  |  |
| 3 | Frich et al, 2006[62] |  | ✓ |  |  | ✓ |  |  | ✓ |  |  |  |  |  | ✓ |  |  |  |  |  |  |
| 3 | Frich et al, 2007[63] |  |  |  |  |  |  |  | ✓ |  |  |  |  | ✓ | ✓ |  |  |  |  |  |  |
| 3 | Frich et al, 2007[24] |  | ✓ |  | ✓ | ✓ | ✓ |  | ✓ | ✓ | ✓ |  |  | ✓ | ✓ |  |  | ✓ | ✓ | ✓ |  |
| 4 | Hallowell et al, 2017[51] |  |  | ✓ |  | ✓ | ✓ |  |  | ✓ | ✓ | ✓ |  |  | ✓ |  | ✓ |  |  |  | ✓ |
| 4 | Jenkins et al, 2013[53] |  |  | ✓ |  | ✓ | ✓ |  | ✓ | ✓ | ✓ |  |  | ✓ |  |  | ✓ |  |  |  |  |
| 4 | Jenkins et al, 2013[52] | ✓ | ✓ |  |  |  |  |  |  |  |  |  |  | ✓ |  |  |  |  |  |  |  |
| 5 | Hardcastle et al, 2015[23] | ✓ | ✓ | ✓ | ✓ | ✓ | ✓ | ✓ |  |  | ✓ |  |  | ✓ | ✓ |  | ✓ | ✓ | ✓ | ✓ | ✓ |
| 6 | Hollands et al, 2012[54] |  |  |  |  | ✓ |  |  | ✓ | ✓ |  |  |  |  |  |  | ✓ |  |  | ✓ |  |
| 7 | Hollman et al, 2004[55] |  | ✓ |  | ✓ | ✓ | ✓ |  |  |  |  |  |  | ✓ |  | ✓ |  |  |  |  |  |
| 8 | Keenan et al, 2018[56] |  | ✓ |  | ✓ | ✓ | ✓ |  |  |  | ✓ | ✓ |  | ✓ | ✓ | ✓ |  | ✓ |  | ✓ | ✓ |
| 9 | Kirkegaard et al, 2014[57] |  | ✓ |  | ✓ | ✓ | ✓ | ✓ | ✓ |  |  | ✓ | ✓ | ✓ | ✓ |  |  |  | ✓ | ✓ |  |
| 10 | Mackie et al, 2015[58] |  | ✓ |  | ✓ | ✓ | ✓ | ✓ |  |  | ✓ | ✓ | ✓ | ✓ | ✓ | ✓ |  |  |  | ✓ |  |
| 10 | Sliwinski et al, 2017[25] | ✓ |  |  |  | ✓ |  |  |  |  | ✓ | ✓ | ✓ | ✓ | ✓ |  |  |  |  | ✓ |  |
| 11 | Meulenkamp et al, 2008[59] | ✓ | ✓ |  | ✓ | ✓ | ✓ |  |  |  | ✓ | ✓ | ✓ | ✓ | ✓ |  | ✓ | ✓ | ✓ | ✓ | ✓ |
| 12 | Mortensen et al, 2008[60] |  |  |  | ✓ | ✓ |  | ✓ | ✓ |  | ✓ |  |  | ✓ | ✓ | ✓ |  |  |  |  |  |
| 13 | Urke, 2016[47] | ✓ | ✓ | ✓ | ✓ | ✓ | ✓ | ✓ |  |  |  | ✓ | ✓ | ✓ | ✓ |  | ✓ | ✓ | ✓ | ✓ |  |

Descriptive themes

Continued

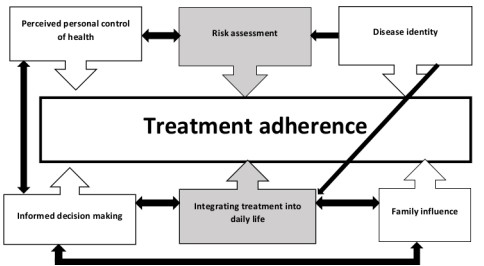

**Figure 2** Thematic schema illustrating influence of analytical themes on treatment adherence.

from being under the care of their parents to assuming responsibility for their behaviours.[25 47]

Growing up surrounded by family members following treatment recommendations and establishing healthy behaviours from a young age was found to instil lifelong habits in individuals.[25 47 48 56 58 59] Those who had grown up from a young age alongside diagnosed family members spoke of their condition and its treatment as something that had become 'normalised'[47] as it was all they had ever known.[25 48 56 58 59] Those who had parents who had bad experiences of medication were apprehensive about taking tablets,[58] but for many it led to the view that taking medication was ordinary[56] and not a 'big deal'.[58]

Two enablers to treatment adherence were identified from these findings: commencement of treatment from a young age and having other family members following similar treatment regimes.

### Informed decision-making

Individuals lacked an in-depth understanding of their disease and its treatment,[23–25 47–51 56–59 61] with many having 'unanswered questions'[49] and requesting more information.[25 49–51] Misconceptions and false information regarding the role of treatment for FH were prevalent[24 25 47–49 51 56–59 61]: 'you can actually eat a lot of fat and the medicine takes care of it.'[23] Individuals were worried about the longer term impact of statin therapy on their and their children's health[49 58] as 'it is a recent drug, and you don't know what the long term effect could be.'[56] Lived experience of side effects were reported by some individuals[49 58 60] and many more were fearful of developing them in the future[55 56 58] as 'many others have severe side effects from what I'm taking'.[60] This incorrect and/or inadequate knowledge of treatment advice and concerns over the short-term and long-term use of lipid lowering medication were identified as barriers to treatment adherence.

Individuals frequently mentioned their encounters with healthcare professionals (HCPs),[23 24 46–48 50 52 53 56 57 59 60] viewing them as playing a 'big role'[25] in their 'team approach'[58] to the management of their FH. Regardless of whether individuals recalled these encounters in a positive[24 25 47 48 50 56 58] or negative[24 46 47 56 60] light, these interactions and relationships with HCPs influenced their understanding of FH and its treatment.

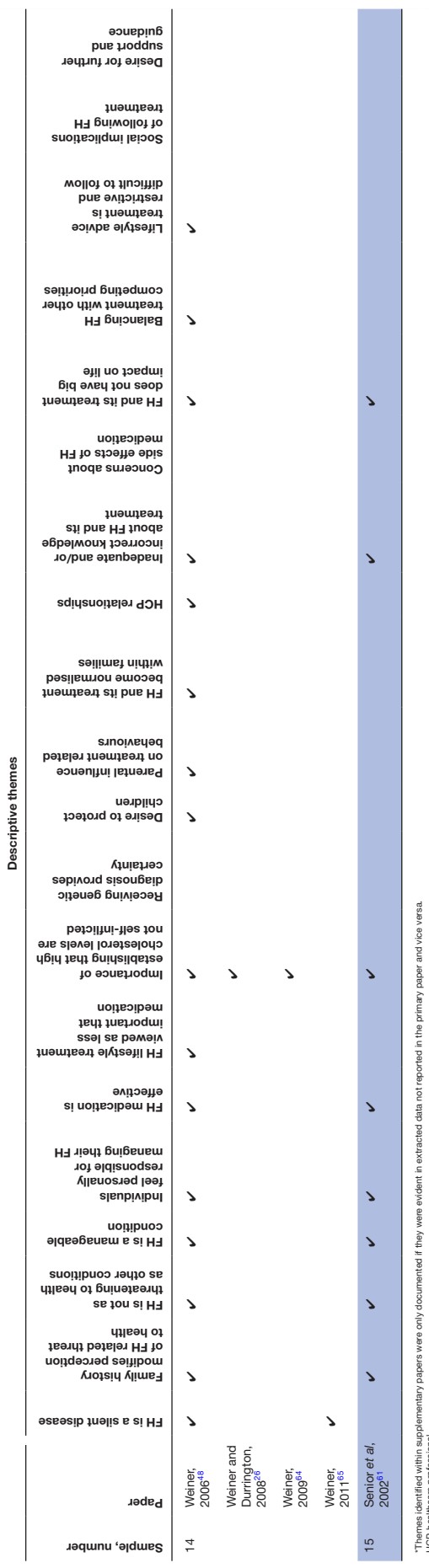

**Table 3** Continued

| Sample, number | Paper | FH is a silent disease | Family history modifies perception of FH related threat to health | FH is not as threatening to health as other conditions | FH is a manageable condition | Individuals feel personally responsible for managing their FH | FH medication is effective | FH lifestyle treatment viewed as less important than medication | Importance of establishing that high cholesterol levels are not self-inflicted | Receiving genetic diagnosis provides certainty | Desire to protect children | Parental influence on treatment related behaviours | FH and its treatment become normalised within families | HCP relationships | Inadequate and/or incorrect knowledge about FH and its treatment | Concerns about side effects of FH medication | FH and its treatment does not have big impact on life | Balancing FH treatment with other competing priorities | Lifestyle advice treatment is restrictive and difficult to follow | Social implications of following FH treatment | Desire for further support and guidance |
|---|---|---|---|---|---|---|---|---|---|---|---|---|---|---|---|---|---|---|---|---|---|
| | | | | | | | | | **Descriptive themes** | | | | | | | | | | | | |
| 14 | Weiner, 2006[48] | ✓ | ✓ | ✓ | ✓ | ✓ | ✓ | ✓ | ✓ | | ✓ | ✓ | ✓ | ✓ | ✓ | | ✓ | ✓ | ✓ | | |
| | Weiner and Durrington, 2008[26] | | | | | | | | ✓ | | | | | | | | | | | | |
| | Weiner, 2009[64] | | | | | | | | ✓ | | | | | | | | | | | | |
| | Weiner, 2011[65] | ✓ | | | | | | | | | | | | | | | | | | | |
| 15 | Senior et al, 2002[61] | | ✓ | ✓ | ✓ | ✓ | ✓ | | ✓ | | | | | ✓ | | | ✓ | | | | |

*Themes identified within supplementary papers were only documented if they were evident in extracted data not reported in the primary paper and vice versa.
HCP, healthcare professional.

**Table 4** Identified enablers and barriers to treatment adherence

| Enablers | Barriers |
| --- | --- |
| Other family members following treatment regime | Mismatch between perceived and actual risk |
| Commencement of treatment from a young age | Concerns over the use of lipid lowering medication |
| Parental responsibility to care for children | Prioritisation of medication over lifestyle treatment |
| Confidence in ability to successfully self-manage their condition | Lifestyle treatment is difficult to comply with |
| Receiving formal diagnosis of FH | Prioritisation of other life events |
| Practical resources and support for following lifestyle treatment | Inadequate and/or incorrect knowledge of treatment advice |
| A positive relationship with healthcare professional | |

FH, familial hypercholesterolaemia.

### Integrating treatment into daily life

Individuals did not feel they had to make many changes to their everyday life as a result of their diagnosis.[23 47–49 51 54 61] Their disease did not hinder them from 'living the life they wanted'[47] or require consideration when making life decisions[23 47 49 54 61] such as having children.[48 51] However, when faced with other commitments, such as family and career obligations, individuals found it more difficult.[23 25 47 49 54 60 62] During these periods, individuals tended to be less focused on managing their disease viewing it as something they could pick up again when they had more time and energy.[23 25 47 56 62] This prioritisation of other life events over the self-management of condition was identified as a barrier to treatment adherence.

The treatment recommendations were perceived to be simple to follow and to have little impact on their quality of life (QOL).[23 47–49 51 53–56 61] However, this perception is in stark contrast to the actual lived experiences of following treatment—especially the lifestyle recommendations. Dietary advice was perceived to be restrictive and interpreted by individuals to mean they could not eat their favourite foods[24 25 47 48 57 59] or enjoy social occasions[24 25 54 57 59 60]: 'I won't bother eating food I don't like, just to follow a certain diet'.[47] Additionally, individuals were concerned about the opinions of their peers in social situations in which they felt they had to make certain dietary choices.[25 47 48 59 60] These findings were prominent among younger individuals.[25 47 59] As a result, the dietary advice was the 'most difficult aspect'[49] of treatment, with many reporting they struggled to follow them at all times.[23–25 47 48 57 59 60] This finding of dietary advice being perceived as difficult to follow was identified as a barrier to adherence.

Reflective of the difficulties faced when trying to follow treatment guidelines, individuals expressed a need for additional information[23 49 50 56] and 'guidelines in order to help you start that change'.[25] Some sought additional information from their HCPs,[23 25 49 50 56] while others called for practical advice and educational resources,[25 49 50 56] as 'everyone knows the theory, but putting it to practice is quite hard'.[23] From this, practical resources and support for following lifestyle treatment advice was identified as an enabler to treatment adherence.

## DISCUSSION

This synthesis has produced new insights into the factors influencing treatment adherence in FH which have implications for clinical practice and future research.

We found that individuals did not perceive FH as a threat to their health except in those who had experienced symptoms of CVD or had a family history of FH-related CVD, as previously reported by others.[66–69] This low perception of risk may be the result of the disease being relatively symptomless and the adverse consequences too far in the future to comprehend. This idea is reinforced by studies reporting heightened perceived risk among older individuals[70] and young adults perceiving their health to be average or above that of the general population.[16] The minimal threat to health may explain the findings that being diagnosed with FH does not increase psychosocial dysfunction in children,[71 72] nor negatively impact on self-reported QOL or rates of depression and anxiety in adults.[73–76] While these findings are positive, individuals who do not view their disease as a serious threat may be less motivated to adhere to treatment, which may explain the findings of higher self-reported medication adherence in older individuals[77 78] and high non-adherence rates in individuals under 36 years.[79] These findings are concerning as individuals who do not adhere fully to treatment have been found to have higher levels of LDL-C.[77 79 80] Furthermore, while treatment has substantially reduced the risk of CVD, individuals still remain at a higher risk than the general population.[9 81 82] This may be a consequence of LDL-C targets not being met by large numbers of treated adults[15 16 79 80 83] and children[84 85] and/or the presence of other risk factors independently associated with CVD.[86 87]

Our findings suggest this low-risk perception may be mediated by beliefs that the risks are avoidable through effective treatment, in line with previous research.[16 66 72 88] These beliefs have been found to positively influence attitudes towards medication, increasing self-reported intentions to comply with medication[19] and rates of adherence.[89] However, individuals' attitudes toward treatment behaviours may have a greater influence on their intention to engage in treatment than their beliefs.[18] Our findings of negative attitudes towards

certain aspects of treatment are therefore important to explore. We found individuals to perceive dietary recommendations as restrictive and impacting on their QOL, as have others.[72 90] Some also believed they were unnecessary if taking medication, likely explaining low uptakes of lifestyle treatment compared with medication.[66 91] We also found negative attitudes towards medication due to side effects and anxieties about long-term safety, similar to others.[16 83 92] In contrast to these studies, we found anxiety about the development of side effects and complications of long-term use to be more prevalent than lived experience of side effects. These negative attitudes are surprising as the dietary recommendations do not differ substantially from those for the general population and the safety and tolerability of statins have been demonstrated in adults[93] and children.[94–96]

Our finding of widespread inadequate knowledge of the treatment recommendations may explain the negative attitudes. It has been reported previously that awareness of the role of PA in treatment is low,[97] and while individuals are mindful of the need for dietary treatment, little is known about the depth of this knowledge.[72 90 97] This finding may be the result of the inconsistency in treatment advice provided with many not receiving the recommended lifestyle advice[91 98 99] or medication treatment[83 85 91 98 100 101] and for those that do, it is often not provided by HCPs with specialist FH knowledge.[91 99] As a result, we found many individuals are left wanting more information about treatment, in line with previous research.[91 97] This is concerning as many report using the internet to search for such information[91] which cannot be easily regulated and may be fuelling our further finding of a high prevalence of incorrect knowledge. Furthermore, individuals may be falsely interpreting negative media coverage of statin medication[102] to be relevant to their condition. This may be negatively influencing adherence to treatment as concerns about general medication overuse have been found to be heavily influential in shaping attitudes towards FH medication[19] Ensuring individuals have a comprehensive and factually correct understanding of the treatment recommendations is therefore essential to optimise adherence.

As this synthesis highlighted that parents take responsibility for their childs' treatment, it is important to ensure they are knowledgeable about the recommendations to help their children develop healthy habits from a young age. Previous research has found that children who follow dietary guidelines from a young age have more positive attitudes towards this aspect of treatment[71] and have improved dietary intakes in childhood[103–105] which are maintained into young adulthood.[106] Furthermore, forgetfulness is frequently reported as a reason for medication non-adherence[16 72 77 78 80 92] and starting treatment at a young age may help overcome this by instilling a routine, as found by others.[107] It is also important to ensure that when individuals reach an age where they become responsible for their own care, they themselves are equipped with the relevant knowledge

to continue to make informed decisions. While there were insufficient data to draw conclusions about best practice for this age group, it appears that transitioning from living at home, adjusting to new routines and prioritising other things in life are common barriers to be targeted.[25 47]

Our findings also highlight the importance of receiving a genetic confirmation of FH. Receiving a medical diagnosis empowered individuals to take control of their condition, providing motivation to continue or commence medication and lifestyle treatments. The positive influence of diagnosis on medication efficacy beliefs and adherence has been reported in previous research.[67 68 108 109] However, in contrast to our findings, it has been reported that positive genetic results have either no effect[68] or weaken beliefs[108] regarding the efficacy of lifestyle treatment. However, in both cases the changes in beliefs did not have a negative impact on their actual behaviours. Given our further finding that individuals find medical diagnosis useful in social situations, a common identified barrier to adhering to dietary recommendations, it may be that genetic diagnosis exerts positive effect on adherence beyond its influence of illness and treatment beliefs.

### Strengths and limitations

Our thematic synthesis adhered to ENTREQ guidelines and used transparent and robust methodology. The comprehensive search strategy, involvement of more than one researcher at each stage of analysis, input from clinicians to corroborate the interpretation of the results and detailed appraisal of the included studies strengthen our findings. The analytical themes generated were produced from descriptive themes that were each evident across a large number of the included papers. The synthesis included data from 264 individuals with FH and 13 family members across 8 countries, encompassing a wide range of ages, duration of diagnoses, primary and secondary CVD prevention and regional differences in healthcare provision. However, all individuals were from developed countries, the majority had high education levels and there were few from ethnic minority groups. This may limit the generalisability of the findings to all individuals with FH. Furthermore, the majority were recruited from lipid clinics and their beliefs may not reflect those opting out of treatment for their condition. Lastly, there were insufficient papers to explore if the factors influencing treatment adherence differ between adults and children with FH and care should be taken when extrapolating results to younger individuals.

### Implications for clinical practice

We have identified seven enablers and six barriers to treatment adherence (table 4) to be considered by any HCP delivering advice to individuals with FH and have produced the following 12 suggestions for clinical practice:

1. Ensure individuals are aware of the risk to their health, without instilling fear through emphasising the effectiveness of medical and lifestyle treatment.
2. Where possible, ensure all individuals receive genetic confirmation of their condition.
3. Communicate that despite the asymptomatic nature of the condition, adhering to treatment from a young age will deliver the greatest benefits to health.
4. Discuss medication within an FH context, emphasising its necessity and distinguishing it from the use of medication in treatment of other causes of high cholesterol.
5. Provide reassurance that medication is safe and side effects uncommon, with reference to relevant clinical guidelines indicating their safety for use by children highlighted to parents.
6. Inform patients that side effects are specific to each type of medication and encourage discussion of any problems so alternative medications can be offered.
7. Communicate dietary advice as being a lifestyle change rather than a restrictive diet with advice tailored to the individual needs and preferences of each individual.
8. Ensure individuals have a factually correct understanding of the dietary recommendations and provide credible resources individuals can access if they require further support or guidance.
9. The benefits of adhering to lifestyle treatment for management of their disease and their overall well-being should be revisited at each clinic appointment.
10. Treatment should begin early, with parents advised that prior to medication, dietary recommendations can be followed from the age of 5. Non-affected family members can also be encouraged to follow guidelines, facilitating a family-based approach to aid adherence.
11. Treatment advice to be provided in family-based clinics if possible, or ensure adult and paediatric services are closely linked.
12. Adolescent patients to be offered opportunity to transition to an adult clinic between the ages of 16 and 18 to take responsibility for their own treatment before they leave home.

### Comparison with treatment adherence in similar medical conditions

The limited literature regarding treatment adherence in FH makes comparison of findings with the present synthesis difficult. However, extensive research has been conducted into treatment adherence for other chronic conditions which are also asymptomatic in the early stages such as hypertension, high cholesterol from non-genetic conditions and type 2 diabetes mellitus, for which treatment adherence rates are also low.[110 111] While it is beyond the scope of this review to compare and contrast the findings in detail, overall the enablers and barriers were similar to those found to exist for individuals following treatment for these similar conditions. For example,

negative perceptions of medication, beliefs that treatment is not necessary due to lack of symptoms, medication side effects and a lack of knowledge about treatment and/or disease were identified as barriers to adherence for those advised treatment to manage risk factors for the primary and secondary prevention of CVD.[112–114] Furthermore, similar findings have been reported in individuals with type 2 diabetes mellitus.[115–117] A unique finding of the present synthesis, however, was that starting treatment from a young age and being surrounded by other family members following treatment facilitates adherence. This is reflective of the genetic inheritance pattern in which an individual will always have one affected parent, which is uncommon in other chronic conditions. Although support from family members, and the involvement of parents, has been identified as an enabler to treatment adherence for individuals with type 2 diabetes mellitus,[115 118 119] the adherence behaviours that parents with FH model to family members are of particular importance in the treatment of FH.

### Future research

With treatment most effective when started at a young age,[6 10 85] and our findings of a positive effect on later life adherence, further qualitative research exploring the perspectives of children is required to allow HCPs to tailor advice to support maximal adherence during this crucial period. The findings of widespread inadequate and/or incorrect knowledge of the treatment recommendations warrant investigation into what advice is being given and by whom. As individuals who have self-selected to receive treatment have concerns about medication, it is likely that there are many individuals opting not to receive treatment for themselves or their child due to these concerns. Future research is needed to explore their perceptions to develop effective interventions that could encourage them to seek treatment.

### CONCLUSIONS

This qualitative evidence synthesis has systematically reviewed and synthesised the available evidence concerning the experiences and beliefs of individuals with FH regarding their condition and its treatment. It has uncovered several enablers and barriers that are to be used in clinical practice to facilitate optimal treatment adherence in this high-risk clinical population group. It has also highlighted significant research gaps which need to be addressed to gain a more comprehensive understanding of how these individuals can be supported to adhere to lifelong treatment.

**Author affiliations**
[1]The National Institute for Health Research (NIHR), Bristol Biomedical Research Centre (BRC), Nutrition theme, University Hospitals Bristol NHS Foundation Trust and the University of Bristol, Bristol, UK
[2]Psychology Department, Bath Spa University, Bath, UK
[3]Department for Health, University of Bath, Bath, UK

[4]Department of Clinical Biochemistry, University Hospitals Bristol NHS Foundation Trust, Bristol, UK
[5]Population Health Science, Bristol Medical School, University of Bristol, Bristol, UK

**Acknowledgements** We would like to thank Catherine Borwick (Research Engagement Librarian) and Dr Alison Gregory (Research Fellow) at The University of Bristol for their expertise and assistance in the development of this evidence synthesis. We would also like to thank Dr Paul Downie (Consultant at University Hospitals Bristol and Royal United Hospitals Bath NHS Foundation Trusts) for his input into translating the findings into clinical recommendations.

**Contributors** FK and RP devised and carried out the search strategy. FK, RP, FEL and JPHS carried out the study screening and selection stage. FK and JC carried out the study characteristic extraction stage. FK and AS carried out the results data extraction, quality appraisal, data analysis and interpretation. EW also carried out the data analysis and interpretation stages. AH contributed to the development and presentation of the qualitative methodology and results. FK, JPHS and GB translated findings into clinical implications. FK prepared the manuscript. All authors reviewed the manuscript and approved the final version.

**Funding** This evidence synthesis is funded by the National Institute for Health Research NIHR Bristol Biomedical Research Centre (Nutrition theme) at University Hospitals Bristol NHS Foundation Trust and The University of Bristol.

**Competing interests** None declared.

**Patient consent for publication** Not required.

**Provenance and peer review** Not commissioned; externally peer reviewed.

**Data availability statement** All data relevant to the study are included in the article or uploaded as supplementary information.

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
