## [Reviewer comments · BMJ Open]

ARTICLE DETAILS

TITLE (PROVISIONAL)	Enablers and barriers to treatment adherence in heterozygous familial hypercholesterolaemia: A qualitative evidence synthesis
AUTHORS	Kinnear, Fiona; Wainwright, Elaine; Perry, Rachel; Lithander, Fiona; Bayly, Graham; Huntley, Alyson; Cox, Jennifer; Shield, Julian; Searle, Aidan

VERSION 1 - REVIEW

REVIEWER	David Armstrong King's College London, UK
REVIEW RETURNED	12-Mar-2019

GENERAL COMMENTS	This paper reports a synthesis of qualitative research into the experience of heterogenous familial hypercholesterolaemia drawing out enablers and barriers to adherence to medication and lifestyle change. This is a well conducted and clearly written study. My only concern is what might be termed the framing of the research question. At the moment, most of the introduction describes FH and its management leading to the conclusion that adherence is an important factor. The alternative frame would be to locate the paper in the adherence literature (and discuss this in the introductory section) before asking the question whether FH is different. in other words I' not sure tthe authors have addressed the most interesting (and important) question. There are lots of conditions nowadays that are less 'diseases' and more 'risk factors' that need constant monitoring and management. They tend to be characterised by lack of symptoms and require the patient to engage with the concept of risk in determining the significance of their condition. Diabetes and hypertension spring to mind. I can see that for all these 'risk factor diseases' adherence might be a particular problem: but the important question surely is whether FH is any different or can we have a simple read across from other similar conditions? Healthcare professionals managing these long-term conditions really need to be aware of general adherence enablers/barriers rather than learning specific ones for each disease, especially as they are likely to be very similar (if not exactly the same). Posing the question this way might save us from having to read a different paper for every condition each reporting the same thing. So, when
---

	I read the list of barriers and enablers this paper would they equally, say, apply to diabetes? Or is FH exceptional in any way? Given this is a nicely written and presented paper it would seem little harsh to demand a complete reorientation as suggested above but least some reference to the existing adherence literature and how FH barriers and enablers might differ could be addressed in the discussion.
--	--

REVIEWER	Leo Akiyamen University of Toronto, Ontario, Canada
REVIEW RETURNED	20-Mar-2019

GENERAL COMMENTS	Strengths: In this present qualitative evidence synthesis, Ms. Kinner and colleagues evaluate barriers and enablers to treatment adherence in heterozygous FH. This is an important and timely question. And, among this study's many strengths are its rigorous and transparently reported protocol, its use of evidence-based and best practices (CASP, ENTREQ, PROSPERO) in qualitative evidence synthesis, and its provision of an accurate assessment of current evidence and suggestions for future investigations. Notwithstanding, I believe this manuscript could benefit from implementing the following amendments. MAJOR COMMENTS Introduction: (1) The entire second paragraph could be omitted or truncated entirely. (2) Entire introduction can be shortened. (2) Tenses switch in objectives section. Results: Would prefer if described in written text, the number of records screened at title/abstract and full text stages. Also, should include reasons for exclusion at full-text stage in the full text. Place Country and Recruitment Setting in separate columns in Table 1. Discussion: My greatest critique of the discussion is that it could benefit from significant truncation. Very much of its current state doesn't really seem to (1) clearly delineate the major findings of this paper, (2) really underscore the significance of these findings, or (3) describe other conclusions that are able to readily extrapolate from the findings of this systematic review. Essentially, a lot of the importance is lost in its verbosity. For example, the significance of individual barriers and enablers to treatment or these in toto is lost in the mass of text. Paragraph 9: A recent systematic review and meta-analysis actually suggests that molecular diagnoses of FH increased adherence to treatment (see: Genetic testing for familial hypercholesterolemia: Impact on diagnosis, treatment and cardiovascular risk).
---

	I suggest distilling their implications for clinical practice into either an itemized list, summative figure or infographic. A visual representation of these would likely assist clinicians with implementation in the future. MINOR COMMENTS Typo (page 5 line 42): aswell Typo (page 5 line 48): ...data from the primary paper PhD theses were... Typo (page 5 line 59): NVivo* Typo (Table 2, column 3, row 4): Huntington's Unfamiliar with the term "thickness" (page 6 lines 15/17). Do we mean "thick description"?: https://academic.oup.com/ndt/article/31/6/897/1751656
--	---

VERSION 1 – AUTHOR RESPONSE

Response to Reviewer 1

This is a well conducted and clearly written study. My only concern is what might be termed the framing of the research question. At the moment, most of the introduction describes FH and its management leading to the conclusion that adherence is an important factor. The alternative frame would be to locate the paper in the adherence literature (and discuss this in the introductory section) before asking the question whether FH is different. In other words I'm not sure if the authors have addressed the most interesting (and important) question. There are lots of conditions nowadays that are less 'diseases' and more 'risk factors' that need constant monitoring and management. They tend to be characterised by lack of symptoms and require the patient to engage with the concept of risk in determining the significance of their condition. Diabetes and hypertension spring to mind. I can see that for all these 'risk factor diseases' adherence might be a particular problem: but the important question surely is whether FH is any different or can we have a simple read across from other similar conditions? Healthcare professionals managing these long-term conditions really need to be aware of general adherence enablers/barriers rather than learning specific ones for each disease, especially as they are likely to be very similar (if not exactly the same). Posing the question this way might save us from having to read a different paper for every condition each reporting the same thing. So, when I read the list of barriers and enablers this paper would they equally, say, apply to diabetes? Or is FH exceptional in any way? Given this is a nicely written and presented paper it would seem little harsh to demand a complete reorientation as suggested above but at least some reference to the existing adherence literature and how FH barriers and enablers might differ could be addressed in the discussion.

Many thanks for your positive comments about our research and for your suggestion to think of the wider application of the findings. We agree that your suggested approach to place the question within the existing adherence literature would be a valid and interesting approach. However, in line with the thoughts of the second reviewer, we feel that focussing on FH is warranted given its high prevalence (1 in 250 individuals) and the current effort across several countries to increase the diagnostic rate which will substantially increase the number of individuals requiring treatment in the near future. FH was specifically referenced in the 2019 NHS 10-year plan in which the NHS stated it aimed to increase access to genetic testing for FH. (NHS, 2019) It stated that NHS England are aiming to identify 25% of people with FH in the next 25 years which will see 37,500 newly diagnosed patients requiring treatment.

However, we agree that referencing to the existing literature would be a good addition to the discussion. We have included this within the 'implications for clinical practice', where we have commented on the transferability of the findings to other similar chronic conditions: "Some of our findings and clinical implications may be relevant to other chronic diseases which are asymptomatic in the early stages such as hypertension and Type II diabetes, for which treatment adherence rates are also low.110 111"

References

NHS (2019) The NHS Long Term Plan. Online. Available at: <https://www.longtermplan.nhs.uk/publication/nhs-long-term-plan/> (Accessed 25/04/2019)

Response to Reviewer 2

(1) The entire second paragraph could be omitted or truncated entirely. The second paragraph has been omitted.

(2) Entire introduction can be shortened. With the omission of the second paragraph, the introduction is now only three paragraphs. We feel all the included text is required to provide background and rationale for the research study described in this manuscript, especially for those readers who may not be familiar with qualitative syntheses.

(3) Tenses switch in objectives section. We have reviewed tenses throughout the manuscript and adjusted these accordingly. However we felt the tenses within the objectives were already consistent with the rest of the manuscript, and these have remained unchanged.

(4) Would prefer if described in written text, the number of records screened at title/abstract and full text stages. Also, should include reasons for exclusion at full-text stage in the full text. This information has been added in- under the 'searches' heading in the results section on page 8, lines 254-263.

The new text reads: "The titles and abstracts of 990 unique citations identified by the searches were screened, with 50 progressing to screening at the full-text level. Twenty-six papers were excluded at this stage due to: the full text not being available (n=1), no primary qualitative data being presented in the findings (n=6), the study population not having a clinical diagnosis of FH or inability to selectively extract the data from those with a diagnosis in a mixed population (n=16) and data not being relevant to the aims of this review (n=3)."

(5) Place Country and Recruitment Setting in separate columns in Table 1. This information has now been split into separate columns in Table 1.

(6) My greatest critique of the discussion is that it could benefit from significant truncation. Very much of its current state doesn't really seem to clearly delineate the major findings of this paper, really underscore the significance of these findings, or describe other conclusions that are able to readily

extrapolate from the findings of this systematic review. Essentially, a lot of the importance is lost in its verbosity. For example, the significance of individual barriers and enablers to treatment or these in toto is lost in the mass of text. We acknowledge that the discussion was very lengthy and agree that shortening it helps to address the above raised points. We have reduced the word count of the discussion by over 700 words. We have also revised the discussion points to better reflect the significance of the findings to treatment adherence. Please see the revised discussion on pages 22-23, lines 16-414.

(7) Paragraph 9: A recent systematic review and meta-analysis actually suggests that molecular diagnoses of FH increased adherence to treatment (see: Genetic testing for familial hypercholesterolemia: Impact on diagnosis, treatment and cardiovascular risk). Thank you for bringing this recent and important publication to our attention. We have now included it in this part of the discussion and reworded as appropriate. Please see page 23, line 409.

(8) I suggest distilling their implications for clinical practice into either an itemized list, summative figure or infographic. A visual representation of these would likely assist clinicians with implementation in the future. Thank you for this useful suggestion. We have now distilled the implications for clinical practice into an itemised list which is on pages 24-25, lines 699-863. This also helped to further reduce the word count and verbosity of the discussion.

(9) Typo (page 5 line 42): aswell. This has been changed to 'as well'

(10) Typo (page 5 line 48): ...data from the primary paper PhD theses were... We have removed the 'the' preceding the sentence above.

(11) Typo (page 5 line 59): NVivo* Nvivo has been changed to NVivo

(12) Typo (Table 2, column 3, row 4): Huntington's Huntingdon's has been corrected to Huntington's.

(13) Unfamiliar with the term "thickness" (page 6 lines 15/17). Do we mean "thick description"?: <https://academic.oup.com/ndt/article/31/6/897/1751656>. Thank for bringing this useful publication to our attention. It's account of 'thick description' is what we meant when describing the 'thickness' of the data. It better summarises the description than the references we originally included, so we have replaced those references with this one and provided further details of its definition within the main body of text on page 7, lines 238-240: 'Thickness' refers to the depth, scope and context of findings which could influence the transferability and credibility of the results to the wider FH patient population. (Craig et al, 2014)'

VERSION 2 – REVIEW

REVIEWER	David Armstrong King's College London
REVIEW RETURNED	08-May-2019

GENERAL COMMENTS	I'm afraid I still think it extraordinary that a paper about treatment adherence completely fails to summarise or even mention the existing literature on that topic. This is justified (in the authors' response) on the grounds that FH is a relatively common condition and may have its own enablers/barriers to adherence. But hypertension or diabetes, say, are far more common and are areas where adherence has been extensively studied. Is there nothing we can learn from the latter? The only interesting question to my mind is whether FH differs in terms of enablers and barriers to adherence compared to these other conditions. My guess is that there will be few or no differences but I cannot tell from this paper as it fails to inform me of what is already known on this topic. Sorry to be so blunt but otherwise we might spend time and energy re-inventing an adherence literature for every condition when in fact the results will be all the same.
---

REVIEWER	Leo Akioyamen University of Toronto Canada
REVIEW RETURNED	09-May-2019

GENERAL COMMENTS	No additional comments. Excellent work.
---

VERSION 2 – AUTHOR RESPONSE

Response to Reviewer 1

I'm afraid I still think it extraordinary that a paper about treatment adherence completely fails to summarise or even mention the existing literature on that topic. This is justified (in the authors' response) on the grounds that FH is a relatively common condition and may have its own enablers/barriers to adherence. But hypertension or diabetes, say, are far more common and are areas where adherence has been extensively studied. Is there nothing we can learn from the latter? The only interesting question to my mind is whether FH differs in terms of enablers and barriers to adherence compared to these other conditions. My guess is that there will be few or no differences but I cannot tell from this paper as it fails to inform me of what is already known on this topic. Sorry to be so blunt but otherwise we might spend time and energy re-inventing an adherence literature for every condition when in fact the results will be all the same.

Thank you for raising our awareness of the importance of considering the results of the synthesis within the context of more extensively researched medical conditions. We agree this is an important area to explore and understand why the previous revision did not satisfactorily answer your original concern. We still feel that as FH is such a common, yet under researched, condition it warrants

investigation into treatment adherence as a standalone piece of research. Therefore we feel that the discussion should focus upon the implications of the results for treatment provision for individuals with FH. However, we have added in a paragraph to give an overview of how the results compare to the enablers and barriers identified for other conditions. There is much more to be said on the subject, but we feel we would not be able to go into this level of detail without considerably extending the wordcount and/or deviating too far from the original aims of the synthesis. We feel the paragraph does answer your question about whether FH differs to other conditions and hope that you feel it is an adequate summary of what is already known on the topic. We have also added in a sentence to the end of the introduction to state that we intend to compare the results to the literature available for other common conditions.

Sentence in introduction:

'Given the limited literature concerning treatment adherence in FH, the results of this synthesis will also be compared to the results of research investigating treatment adherence in similar medical conditions'

Paragraph in discussion:

'Comparison with treatment adherence in similar medical conditions

The limited literature regarding treatment adherence in FH makes comparison of findings with the present synthesis difficult. However, extensive research has been conducted into treatment adherence for other chronic conditions which are also asymptomatic in the early stages such as hypertension, high cholesterol from non-genetic conditions and type 2 diabetes mellitus, for which treatment adherence rates are also low. (Ramli et al., 2012, Polonsky and Henry, 2016) While it is beyond the scope of this review to compare and contrast the findings in detail, overall the enablers and barriers were similar to those found to exist for individuals following treatment for these similar conditions. For example, negative perceptions of medication, beliefs that treatment is not necessary due to lack of symptoms, medication side effects and a lack of knowledge about treatment and/or disease were identified as barriers to adherence for those advised treatment to manage risk factors for the primary and secondary prevention of CVD. (Leslie et al., 2018, Sud et al., 2005, Alm-Roijer et al., 2004) Furthermore, similar findings have been reported in individuals with type 2 diabetes mellitus. (Tiv et al., 2012, Broadbent et al., 2011, Pollack et al., 2010) A unique finding of the present synthesis, however, was that starting treatment from a young age and being surrounded by other family members following treatment facilitates adherence. This is reflective of the genetic inheritance pattern in which an individual will always have one affected parent, which is uncommon in other chronic conditions. Although support from family members, and the involvement of parents, has been identified as an enabler to treatment adherence for individuals with type 2 diabetes mellitus, (Tiv et al., 2012, Miller and Dimatteo, 2013, Rintala et al., 2013) the adherence behaviours that parents with FH model to family members is of particular importance in the treatment of FH.'

VERSION 3 - REVIEW

REVIEWER	David Armstrong King's College London, UK
REVIEW RETURNED	20-Jun-2019

GENERAL COMMENTS	Thank you to the authors meeting my concerns. The new paragraph in the discussion provides a context for considering the findings of this systematic review. Essentially, it seems that FH follows the adherence pattern of other asymptomatic risk factors (as I suspected) – except for the importance of family member involvement. In a sense, the latter is the main finding of this study and I would have been inclined to stress it more in the abstract and conclusion but I am happy that the paper now places its results into an appropriate context and I would not want to argue for further delay to acceptance.
--